# Microalgae-Based Biorefineries: Challenges and Future Trends to Produce Carbohydrate Enriched Biomass, High-Added Value Products and Bioactive Compounds

**DOI:** 10.3390/biology11081146

**Published:** 2022-07-29

**Authors:** Eugenia J. Olguín, Gloria Sánchez-Galván, Imilla I. Arias-Olguín, Francisco J. Melo, Ricardo E. González-Portela, Lourdes Cruz, Roberto De Philippis, Alessandra Adessi

**Affiliations:** 1Environmental Biotechnology Group, Biotechnological Management of Resources Network, Institute of Ecology (INECOL), Xalapa, Veracruz 91073, Mexico; gloria.sanchez@inecol.mx (G.S.-G.); francisco.melo@inecol.mx (F.J.M.); erik.gonzalez@inecol.mx (R.E.G.-P.); janeeyre3571@gmail.com (L.C.); 2International Centre for Clean Technologies and Sustainable Development, (CITELDES, A.C.), Xalapa, Veracruz 91190, Mexico; iiaolguin@outlook.com; 3Department of Agriculture Food Environment & Forestry (DAGRI), University of Florence, 50144 Florence, Italy; roberto.dephilippis@unifi.it (R.D.P.); alessandra.adessi@unifi.it (A.A.)

**Keywords:** third-generation biorefineries, multi-product biorefineries, circular economy, bioactive compound, mixotrophic cultures, wastewater

## Abstract

**Simple Summary:**

Microalgae-based biorefineries allow the simultaneous production of microalgae biomass enriched in a particular macromolecule and high-added and low-value products if a proper selection of the microalgae species and the cultivation conditions are adequate for the purpose. This review discusses the challenges and future trends related to microalgae-based biorefineries stressing the multi-product approach and the use of raw wastewater or pretreated wastewater to improve the cost-benefit ratio of biomass and products. Emphasis is given to the production of biomass enriched in carbohydrates. Microalgae-bioactive compounds as potential therapeutical and health promoters are also discussed. Future and novel trends following the circular economy strategy are also discussed.

**Abstract:**

Microalgae have demonstrated a large potential in biotechnology as a source of various macromolecules (proteins, carbohydrates, and lipids) and high-added value products (pigments, poly-unsaturated fatty acids, peptides, exo-polysaccharides, etc.). The production of biomass at a large scale becomes more economically feasible when it is part of a biorefinery designed within the circular economy concept. Thus, the aim of this critical review is to highlight and discuss challenges and future trends related to the multi-product microalgae-based biorefineries, including both phototrophic and mixotrophic cultures treating wastewater and the recovery of biomass as a source of valuable macromolecules and high-added and low-value products (biofertilizers and biostimulants). The therapeutic properties of some microalgae-bioactive compounds are also discussed. Novel trends such as the screening of species for antimicrobial compounds, the production of bioplastics using wastewater, the circular economy strategy, and the need for more Life Cycle Assessment studies (LCA) are suggested as some of the future research lines.

## 1. Introduction

Microalgae have been recognized as a very important group of photosynthetic microorganisms for many of their attributes and the roles that they play in nature. Their importance as one of the major groups for CO_2_ sequestration and for releasing oxygen either in marine environments [1], in freshwater bodies [2], and even in arid and semiarid environments [3], has attracted the attention of several research groups for decades. On the other hand, microalgae have been identified as a very versatile and diverse group of organisms with a very high potential for application in biotechnology [1,4] due to their high capacity to produce and accumulate various macromolecules (proteins, carbohydrates, and lipids) with several uses in the food and feed industry, and more recently, as a source of biofuels [5,6]. Furthermore, microalgae are a very important source of several high-added value products such as pigments, PUFAs (poly-unsaturated fatty acids), peptides, exo-polysaccharides, among others, with a clear benefit to human health and nutrition [7,8] (Figure 1).

Despite all of these attributes that indicate that microalgae are a very useful feedstock for biofuels [9], food and feed supplements [10], and high-added value products [11], it has been recognized that several constraints still have to be overcome to make their large-scale production economically feasible, such as low productivity, risk of contamination in outdoor cultures, and high harvesting and downstream processing costs [12]. Thus, one of the strategies to overcome such drawbacks is their production following a “Biorefinery” concept.

A widely used Biorefinery definition is the one launched by the IEA (International Energy Agency) Bioenergy Task 42: “Biorefinery is the sustainable processing of biomass into a spectrum of marketable products (food, feed, materials, chemicals) and energy (fuels, power, heat)” [13], and it has been applied in recent decades to the processing of biomass for production of biofuels and chemical products with the expectation of reduced environmental impact [14,15]. It has drawn the attention of several research groups following different approaches. With the purpose of optimizing the environmental and economic performance of a biorefinery, a useful tool that combines the economic value and the environmental impact (EVEI) has been developed [16].

On the other hand, microalgae have been used for treating wastewater since many decades ago, when the pioneer group led by Prof. William Oswald initiated their use on a large scale in California [17]. Following this approach, research has been oriented toward the design and establishment of integrated systems in which microalgae were used for treating wastewater and producing valuable biomass [18,19,20,21]. Currently, microalgae-based biorefineries have been investigated by several research groups [22,23,24], and in some cases, such a type of biorefinery has integrated the treatment of wastewater and used some of the treated effluents or digestates as a source of nutrients [25,26,27].

It is important to mention that both the integrated systems for treating wastewater with microalgae and the microalgae-based biorefineries treating and using wastewater follow the circular economy concept (Figure 2). In fact, the term circular economy has been described as the efficient utilization of biomass, including waste, for the sustainable production of high-added and low-value products (e.g., food, feed, biomaterials, biofuels, biofertilizers, and biostimulants) with the benefit of lowering the carbon footprint and the valorization of waste materials [28].

The aim of this review is to highlight and discuss the microalgae-based biorefineries, including both phototrophic and mixotrophic cultures treating wastewater and the recovery of biomass as a source of valuable macromolecules and high-added value products. The therapeutic properties of some microalgae-bioactive compounds are also discussed.

## 2. Photoautotrophic Production of High-Added Value Products

Despite the large potential of several microalgae for producing high-added value products, only a few species are produced at a large scale under photoautotrophic conditions: *Arthrospira* sp., *Chlorella* sp., *Dunaliella* sp., *Haematococcus* sp., and *Nannochloropsis* sp. [29,30,31,32,33].

The production of food supplements or specific bioactive compounds gained interest for large-scale commercialization [34], mainly due to the demand in high-income countries. The genus *Arthrospira* sp. is the one which is being produced in the largest quantity (tons/year) and in many countries due to its high content of protein and other well-known high added-value compounds [35]. Earthrise Nutraceuticals^®^ started production of this valuable cyanobacteria in California and Hawaii more than 40 years ago. *Arthrospira* is currently one of the most widely commercially cultivated microalgae in open raceway ponds worldwide, with China and the United States leading the global production, followed by Japan, Taiwan, Thailand, India, and Australia. According to a recent publication from Meticulous Research, the *Arthrospira* market is expected to record a compounded annual growth rate (CAGR) of 13.2% from 2021 to 2028 to reach USD 968.6 million dollars by 2028. In terms of volume, the *Arthrospira* market is expected to register a CAGR of 18.1% from 2021 to 2028 to reach 98,768.5 tons by 2028 [36,37].

The second genus produced ina very large quantity is *Chlorella*, also in various countries. *Dunaliella* sp. is being produced either as whole dried biomass or for the direct production of β-carotene. *Haematococcus* sp. has also been cultivated outdoors for the production of asthaxanthin, a very valuable pigment [35].

On the other hand, the production of pigments is still an attractive field of research, and there are several recent interesting reports dealing with different operational strategies (batch and semi-continuous cultures) and showing variable yield and productivity, depending on each strategy used and the strain of microalgae utilized (Table 1). It is important to note that new information regarding new strains or new types of processes might lead to new commercial large-scale production worldwide.

A computational model to evaluate the techno-economic viability of high-value compound extraction together with the utilization of the extracted biomass for low-cost applications such as bioenergy or biomaterials from microalgae has been recently reported by Resvani et al. [51]. These authors developed models to determine the technical setups, capital requirements, and operating costs of three types of closed photobioreactors (PBR): flat-panel, airlift, and tubular systems, and concluded that flat-panel photobioreactors showed the best overall economic result. They also applied sensitivity analyses to understand the effect of dominant parameter variations on the target price of microalgae and the available high-value compounds. Such analysis showed that the ratio of high-value metabolites is the most crucial factor determining economics and indicated a low-level impact of productivity and cultivation costs on high-priced metabolites.

Concerning the production of phycocyanin (PC), one of the most valuable pigments from cyanobacteria, some drawbacks still need to be overcome to achieve economic feasibility at large-scale production. A recent report [40] has tested three strategies (Figure 3) to design and implement a feasible process: (1) the reduction in the cost of the culture media by modifying the conventional medium composition, using urea as the nitrogen source instead of nitrates, avoiding toxicity by using a fed-batch mode during the cultivation stage; (2) the induction of stress conditions using blue light, aimed at increasing the yield of PC; and finally, (3) the incorporation of a low energy consumption process for harvesting and recovering PC, such as membrane filtration technology. The implementation of such strategies resulted in a high yield of phycocyanin (150 mg PC g^−1^) with a high purity ratio (3.72 ± 0.14).

## 3. Mixotrophic Production of High-Added Value and Low-Value Products Using Wastewater

Following the concept of circular economy and sustainable development, microalgae-based biorefineries treating and using wastewater have also been an important field of research in recent decades [5]. The initial trend was to produce biomass for biofuels production [25,26,27,52], and more recently, the trend is to produce biomass for biofuels and high-added value products [53,54,55]. Within this context, *Arthrospira platensis*, *Haematococcus pluvialis*, *Chlorella sorokiniana*, *Dunaliella* sp., and *Nannochloropsis limnética*, *Nannochloropsis gaditana*, among others, have been cultivated in municipal wastewater [56,57,58,59], industrial wastewater [60,61,62], and agroindustrial wastewater [63,64,65] for the production of some of the most demanded pigments (astaxanthin and β-carotene), as shown in Table 2. There have been estimations that by using this type of mixotrophic culture, the cost of biomass could be less than 5 EUR/kg of biomass [66,67].

Furthermore, in the search to overcome the use of common fertilizers and agrochemicals, microalgae biomass grown in wastewater is becoming very useful to be applied as biofertilizer and biostimulant, achieving soil and crop enrichment [68,69,70,71].

Although there are several advantages to using wastewater for the production of high-added value products, there are also limitations to this kind of strategy. A list of such advantages and limitations which offer future trend research opportunities is provided in Table 3.

## 4. Production of Microalgae Biomass Enriched in Carbohydrates

The photosynthetic metabolism of microalgae, as well as the metabolism of carbohydrates and lipids, are similar to those of terrestrial plants. However, the photosynthetic efficiency of microalgae is in the range of 6–10%, exceeding the 1–2% that of terrestrial plants [85]. Some authors mention that in most microalgae, starch accumulates as a primary source of energy and carbon reserve, while lipids serve as secondary storage [86]. Carbohydrate productivity in microalgae is usually higher than the lipid one, as the accumulation of lipids requires intense stress, whereas carbohydrate production is “easily” achieved by photosynthesis through the Calvin cycle and is markedly increased by nitrogen deficiency [87].

The contents of lipids, carbohydrates, and proteins depend on the type of microalgae and the environmental and nutritional conditions applied to the culture [88,89,90]. The ability of microalgae to produce energy-rich compounds, such as lipids and carbohydrates, has led them to be considered one of the most important raw materials used in the production of third-generation biofuels [88].

This review has focused on the production of carbohydrate-enriched biomass since this type of biomass has several applications in the production of bioethanol as well as in the production of valuable exopolysaccharides and bioplastics.

### 4.1. Environmental and Nutritional Factors Affecting Microalgae Cultures

The main environmental factors that affect the growth of microalgae in different media have been widely revised and are pH, light intensity, temperature, and salinity, while nutritional factors include the availability and source of nitrogen, carbon, phosphorus, sulfur, and iron [91]. Recently, Daneshvar et al. [92] discussed the critical aspects that influence microalgae cultivation and biomass production, among them the environmental factors; pH, temperature, irradiation, and aeration, which affect not only microalgae growth but also the biochemical composition of microalgal biomass. On the other hand, Bermejo et al. [93] presented an overview of the techniques to evaluate the status of microalgae cultures (cell viability and vitality) through biomarkers; these can indicate, quantitatively, changes in a biological system exposed to different environmental factors. These biomarkers are biomolecules present in microalgae, such as chlorophyll, ATP, lipid oxidation products, DNA, RNA, and caspase-like proteases. The authors mentioned that the techniques used for the determination of such biomolecules must be considered potential and important tools for determining the quality of microalgae cultures.

It is known that the pH suitable for carbohydrate accumulation differs according to the type of microalgae species used. da Silva-Braga et al. [90] studied the effect of medium pH in heterotrophic cultures of *Spirulina sp.* The highest carbohydrate content (59%) and the highest maximum biomass concentration (0.97 g L^−1^) were obtained when they used NaHCO_3_ as the only source of C (Zarrouk medium) and a reduction of NaNO_3_ (0.25 g L^−1^) only at the beginning of the process. It was attributed to the fact that the slightly alkaline pH of the medium kept the HCO_3_ ions soluble, optimizing their availability for the microalgae, unlike what happened with the CO_2_ injections, which acidified the medium, causing less availability of C for the cultures.

### 4.2. The Effect of Irradiance on Carbohydrate Accumulation

It is well known that irradiance is one of the major factors affecting the growth of microalgae. Stress conditions, including irradiance and the overproduction of biofuels (biodiesel and bioethanol) from microalgae, have been extensively discussed previously [94,95]. Furthermore, a very comprehensive kinetic analysis of microalgae cultures developed to provide a guideline for photobioreactor design and process development has also been reported [96].

Rodríguez-Miranda et al. [97] developed a simulation-based methodology to establish the influence of season and culture depth on the one-year irradiance and temperature of the culture, determining the biomass productivity of different microalgae strains. It is expected that such methodology could be used as an easy-to-manage decision-making tool for the optimal design and operation of large-scale microalgae facilities of the most cultivated microalgae strains (*Spirulina platensis, Chlorella vulgaris, Nannochloropsis gaditana, Isochrysis galbana*, and *Scenedesmus almeriensis*).

This section is intended to provide a brief account of the effect of irradiance on carbohydrate accumulation. The combination of suboptimal light intensity and nitrogen deficiency also has a strong effect on the biochemical composition of microalgae. In an early work performed with *Arthrospira* sp. cultivated in a complex medium containing seawater supplemented with digestates from pig waste, two different irradiance levels were tested: 66 and 144 µmol m^−2^ s^−1^ [98]. It was found that biomass was enriched in total Iipids (28.6% dry weight (dw)) when cultures were exposed to the lower irradiance level. On the other hand, the total polysaccharide content (28.41% dw) was significantly higher (*p* < 0.05) at the highest irradiance level. It has also been reported that an optimal light intensity (420 µmol m^−2^ s^−1^) for *S. obliquus* CNW-N cultivation with N deficiency and a continuous CO_2_ supply significantly increased carbohydrate production [89]. Higher values of light intensity above the optimal for growth caused a decrease in biomass productivity and the CO_2_ fixation rate, which were explained as excessive illumination or photoinhibition, respectively. It is important to stress that assessment of the effect of environmental conditions on macromolecular composition is better understood in outdoor cultures. Morillas-España et al. [99] have shown that the macromolecular composition of *Scenedesmus almeriensis* cultivated in outdoors raceways under spring and summer using commercial fertilizers as a source of nutrients, showed no major differences with an average protein, lipid, ash, and carbohydrate content of 37.9, 4.6, 10.8, and 46.7%, respectively.

On the other hand, it has been shown that nitrogen depletion and light intensity were important factors affecting triacylglycerol (TAG)/carbohydrate accumulation in *Neochloris oleoabundans* HK-129 cultures [100]. TAG/carbohydrate (feedstock in biofuel production) production was significantly increased when using a two-stage strategy; rapid microalgal growth under nutrient-rich conditions (stage 1), followed by a culture subjected to a period of moderate N depletion, high light intensity, and suitable Fe concentration. Maximum biomass concentration was achieved at 200 µmol m^−2^ s^−1^ light intensity and 0.037 mM Fe.

Lisondro et al. [101] reported the influence of irradiance on the growth and biochemical composition of *Nitzschia aff. pellucid*. They found that the highest content of lipids (~20%) and carbohydrates (~40%) were achieved at high irradiances (≥450 µmol m^−2^ s ^−1^), while the highest protein content (>8%) was found at low irradiances (≤200 µmol m^−2^ s ^−1^).

As a final remark, it should be mentioned that the effect of irradiance on carbohydrate accumulation depends on several factors, mainly the microalgae species, the type of photobioreactor, and the stress conditions provided. It is fundamental that studies should be performed outdoors and during different seasons to provide reliable information for scaling up cultures.

### 4.3. Nutritional Factors (N, P and S) That Affect Carbohydrate Accumulation

Nutrient manipulation can create a stressful environment in microalgae, inducing the biosynthesis of products of interest, such as carbohydrates and lipids. Numerous studies have shown that the cellular composition of microalgae can be affected by nutrient deficiency in the culture [102]. Macronutrients such as nitrogen (N), phosphorus (P), and sulfur (S) control the growth and metabolic activity of microalgae, and their depletion can result in a considerable enhancement of starch and/or lipid content in microalgae cells. However, no high concentrations can be obtained due to the decreased photosynthesis [103]. Yuan et al. [104] have pointed out that macronutrient limitation is more efficient for improving carbohydrate productivity than macronutrient starvation. This is because, in long-term cultivation, nutrient starvation could lead to a high carbohydrate content but affect microalgae growth. Thus, they consider that the most important target is to achieve high carbohydrate productivities instead of high carbohydrate contents.

#### 4.3.1. Nitrogen

Nitrogen concentration regulates cell growth and energy metabolism. Nitrogen deficiency increases the carbohydrate content because, under this condition, proteins are no longer produced, and proteins and peptides already accumulated in the microalgae can be transformed into carbohydrates and lipids [89,102]. There are different nitrogen limitation strategies, such as total limitation, intermediate concentrations of nitrogen, and two-stage cultivation; in the last strategy, microalgae are first cultivated in nitrogen-rich conditions and then transferred to a nitrogen-free BG11 medium [86], and finally, the cultivation time can be prolonged, so that the microalgae consume the nutrients of the medium causing nitrogen starvation in microalgae [105].

Ho et al. [89] assessed some strategies (irradiance intensity and nitrogen starvation) to optimize lipid/carbohydrate productivity of *S. obliquus* CNW-N. The highest carbohydrate productivity was obtained at 420 µmol m^−2^s^−1^ using no N starvation (321.52 mg L^−1^d^−1^). It was observed that when N starvation was applied (140 µmol m^−2^s^−1^), the lipid content increased from 11.5 to 22.4% (maximum value occurred on the fifth day of starvation). The carbohydrate content increased from day 0 (38.3%) to day 1 (46.7%), reaching the maximum value at day 5 of N starvation (51.8%). The maximum carbohydrate productivity was obtained on day 3 of N starvation (383.40 mg L^−1^d^−1^), which was similar to that registered on day 5 (379.40 mg L^−1^d^−1^).

Ho et al. [105], worked with a native strain of *Scenedesmus obliquus* in a two-stage culture with a constant CO_2_ supply. In the first stage, the culture was maintained in a nutrient-rich modified basal medium, and after four days, it was transferred to deionized water. This triggered the carbohydrate content from 25% to 52%, which corresponded to a carbohydrate productivity of 178.40 and 352.90 mg L^−1^d^−1^, respectively. In addition, it was observed that the optimum temperature for this bioprocess was 29 °C, corresponding to the summer temperature of the study region. A predominance in glucose production of 80% of the carbohydrates was also induced.

The effect of nitrogen deficiency on carbohydrate accumulation in *Chlorella vulgaris* FSP-E culture was investigated. Around the fifth day, the culture showed N depletion. During this period the protein content decreased from 58.8% to 20.1%, while the carbohydrate content increased from 13.3% to 54.4%, the maximum value was achieved after three days of nitrogen deficiency. The maximum carbohydrate productivity (687 mg L^−1^d^−1^) was obtained after two-day N starvation. It is necessary to explore the optimum time of microalgae cultivation under nitrogen deficiency conditions to achieve the highest carbohydrate productivity [106].

In a two-stage photoautotrophic culture, the microalga *Chlamydomonas reinhardtii* CC125 improved its biomass productivity from 0.23 g L^−1^ d^−1^ of the *batch* culture (5% CO_2_) to 0.47 g L^−1^ d^−1^ of a repeated fed-batch operation with 70% medium replacement in a first stage. However, no significant differences were observed among the carbohydrate contents obtained (ranging from 7.24 to 8.50%) at the different medium replacement ratios (45, 70, 83, and 91%). In a second stage, the authors studied the growth and biomass composition of *C. reinhardtii* CC125 in liquid Sueoka’s high salt (HS) medium with depleted N and supplied with mixed air containing 0.04 and 5% CO_2_. The increase in carbohydrate content was higher at 0.04 than 5% CO_2_ reaching the maximum value (71%) on day 4 of N starvation. It was concluded that two-stage culture is an efficient strategy to improve carbohydrate production in microalgae [107].

#### 4.3.2. Phosphorus

Phosphorus regulates the cell energy metabolism since it participates in photosynthetic respiration, nitrate adsorption, signaling, phosphorylation, and the dephosphorylation of proteins. P depletion can lead to the accumulation of starch or carbohydrate in microalgae cells [108]. Yao et al. [109] assessed the cell growth and starch accumulation in *Tetraselmis subcordiformis* under extracellular P-limitation with initial cell densities (ICD) of 1.5, 3.0, 6.0, and 9.0 × 10^6^ cells mL^−1^. They found the maximum starch content (44.1%) in the lowest ICD culture followed by the value observed at 3.0 × 10^6^ cells mL^−1^ (42.2%), while the starch contents obtained at higher ICD (6.0 and 9.0 × 10^6^ cells mL^−1^) were significantly lower (28.7 and 24.8%, respectively). The maximum starch concentration (1.6 g L^−1^) and starch productivity (0.30 g L^−1^ d^−1^) were obtained with the ICD of 3.0 × 10^6^ cells mL^−1^. The authors mentioned that an appropriate ICD should be used to regulate the intracellular P concentration and maintain adequate photosynthetic activity to achieve the highest starch productivity, along with biomass and starch concentration.

#### 4.3.3. Sulfur

Sulfur synchronizes photorespiration, electron transport, and starch synthesis and also controls the redox state of the cells. Sulfur starvation can provoke the accumulation of carbohydrates and TAGs [108]. In fact, sulfur starvation or limitation can be most effective for carbohydrate accumulation than nitrogen and phosphorus starvation [104].

Different doses of sulfur were experimented on the microalga *Scenedesmus acuminatus,* because, in theory, it would affect photosynthesis, respiration, N assimilation, and carbohydrate metabolism. It was observed that a decrease in sulfur supply on *S. acuminatus* changed N metabolism, affected its uptake, soluble protein content, and free amino acid composition. While sulfur at high doses favored growth and biomass content, the limitation of this nutrient is that it induced only an increase in lipid accumulation [110]. Triphati et al. [108] found an increase (~33%) in the carbohydrate content of the sulfur-limited cells of a novel microalga *Coccomyxa* sp. IITRSTKM4.

### 4.4. Strategies to Stimulate Carbohydrate Concentration

#### 4.4.1. Selection of the Appropriate Microalgae

The selection of appropriate microalgae species is an important factor for the success of the whole production process due to the following reasons [88]:Tolerance to acute stress (especially in closed photobioreactors);To become dominant in relation to contaminating microorganisms;High CO_2_ absorption capacity in photoautotrophic systems (high photosynthetic efficiency);Tolerance to large temperature variations resulting from daily and seasonal cycles;Low nutrients requirement;Potential to obtain high value-added co-products added to the desired products.Presence of short productive cycles;To show auto flocculation, useful for the microalgal biomass recovery stage.

#### 4.4.2. Operating Modes and Cultivation Systems

Microalgae can grow in autotrophic, heterotrophic, and mixotrophic conditions, in which their capacity to fix carbon and, therefore, their biomass, carbohydrate, and lipid yields vary greatly. Microalgae grown in mixotrophic systems can metabolize both organic and inorganic carbon and, at the same time, play a major role in environmental remediation through the treatment and utilization of flue gas and organic wastewater, which gives them significant advantages over other systems. In comparison, the high costs of heterotrophic crops (capital and operational) and the lower biomass yields produced by autotrophic crops represent a major disadvantage for these systems [6]

The mixotrophic cropping strategy offers numerous advantages over photoautotrophic and heterotrophic cropping modes: (1) higher growth rates to produce higher biomass yields; (2) reduced biomass loss during dark hours due to respiration-mediated growth; (3) comparatively prolonged exponential growth phase; (4) great flexibility to switch from heterotrophic to photoautotrophic mode and vice versa; (5) prevention of photooxidative damage caused by accumulated O_2_ in closed photobioreactors; and (6) reduced photoinhibition of substrate uptake [6]. Mixotrophic cultivation is an economically viable process for large-scale production.

Some microalgae and cyanobacteria have exhibited better growth in mixotrophic conditions than with the other two growth modes, e.g., *Chlorella vulgaris, Chlorella regularis, Chlorella sorokiniana, Spirulina platensis, Haematococcus pluvialis,* and *Euglena gracilis* [88].

It has been demonstrated that carbohydrate production of *C. sorokiniana* NIES-2168 could be significantly improved by an acetate-based mixotrophic culture system (using different concentrations and replacement times of the anion in the modified Bold 6N medium) together with different engineering strategies (batch, fed-batch, and semi-continuous). In particular, the application of mixotrophic culture combined with a semi-continuous operation (with 30% replacement of the medium) gave rise to an increase in biomass productivity of 695 mg L^−1^ d^−1^ (the highest value ever reported). However, the highest carbohydrate content obtained (64.8%) was obtained in the fed-batch condition when carbon was added to the medium every 48 h. Glucose constituted up to 80 to 85% of the total carbohydrate, which is a very suitable proportion for alcoholic fermentation [111].

Considering that the biochemical composition of microalgae will be the result of the operating modes and cultivation systems, Table 4 provides several examples for different microalgae species.

Another important issue to discuss in relation to the operating mode and cultivation system is the one related to the recycling of water. It has been reported that 65%–85% of the water required for large-scale cultivation of microalgae can be saved by recycling water after harvesting the biomass [121]. It has also been mentioned that recycling process water can diminish the environmental impact of releasing high salinity water [122]. It is important to mention that water is not always available in the required volumes in desertic regions where the climatic conditions may be adequate for microalgae growth.

An extensive review of the advantages and limitations of the strategy for recycling water after microalgae cultivation or after subjecting microalgae biomass to anaerobic digestion has been published [83]. This author describes a very useful parameter known as the “biomass ratio”. It is the biomass produced using recycled water under the same conditions as the biomass produced in a control medium. Accordingly, a biomass ratio less than 1 would suggest that growth was inhibited or limited by the presence of macro or micronutrients, unassimilated ions, toxic compounds, or undesirable pH in the recycled water.

#### 4.4.3. Use of Wastewater for Microalgae Cultivation

Nzayisenga et al. [123] evaluated the cultivation of a native *Chlorella sp.* isolated from wastewater, which was very efficient in the removal of N (89%) and P (96%). The highest carbohydrate contents were obtained under autotrophic but mostly mixotrophic conditions, as opposed to heterotrophic conditions, where the highest lipid content was presented.

A two-stage cultivation strategy is very promising for wastewater treatment using microalgae. Farooq et al. [124] reported a cultivation system in a two-stage photoautotrophic–photoheterotrophic/mixotrophic mode to maximize lipid productivity of two strains of *Chlorella* sp. grown in brewery wastewater (BWW). *C. vulgaris* (UTEX-265) and *Chlorella* sp. isolated from BWW were used in this study. The results showed that during the first stage of the two-stage photoautotrophic–photoheterotrophic system, the total nitrogen and phosphorous were removed by more than 90% in both strains. When *C. vulgaris* (UTEX-265) was evaluated, the highest biomass concentration (3.20 g L^−1^) and lipid productivity (108 mg L^−1^ d^−1^) were obtained in the two-stage photoautotrophic–photoheterotrophic (10 g L^−1^ glucose). In the case of *Chlorella* sp., similar biomass concentrations were observed (ranging from 2.1–2.74 g L^−1^) in the different media and system types (photoautotrophic: TAP medium, BWW; two-stage photo-mixotrophic: 25% BWW, 3 g L^−1^ glucose; two-stage photoautotrophic–photoheterotrophic: 5 g L^−1^ glucose. The highest lipid productivity (50.23 mg L^−1^d^−1^) was obtained in two-stage photoautotrophic–photoheterotrophic (5 g L^−1^ glucose).

*Scenedesmus obliquus* showed high adaptability to grow in domestic wastewater. By prolonging the culture time, the total carbohydrate content in the biomass was doubled due to the stress produced by ammonia nitrogen deficiency mainly. Based on the high removal efficiency of N and P (80% and 97.7%, respectively) and total carbohydrate content (31.1%), it was concluded that microalgae cultivation is a technically feasible alternative for wastewater treatment and production of microalgal biomass for obtaining biofuels [125].

On the other hand, the feasibility of using diluted digestate from biogas production for carbohydrate production with five promising microalgae strains has been demonstrated. Tan et al. [126] compared to the modified BBM medium, the diluted biogas digestate was attributed to the highest carbohydrate content in the five microalgae studied, among which *C. vulgaris* ESP-6 obtained the highest carbohydrate content of 61.50% and the highest carbohydrate productivity of 388.34 mg L^−1^ d^−1^. In addition, the absence of phosphorus and magnesium, which may be averse to biomass accumulation, was attributed to the earlier timing of carbohydrate accumulation. Magnesium was first recognized and tested as an influencing factor for carbohydrate accumulation.

According to Chokshi et al. [127], raw wastewater from the dairy industry was found to be very promising in the culture of *Acutodesmus dimorphus,* which implies a great saving of time, water, and nutrients. Only four days of cultivation were sufficient to drastically reduce the level of contaminants (C, N, and P in excess) and produce maximum biomass. The biomass contained 25% lipids and 30% carbohydrates, which can be converted into biofuels. The cultivation of microalgae in raw industrial effluents has two purposes: wastewater treatment and sustainable production of a high amount of biomass.

It has also been demonstrated that mixotrophic cultures of *Chlorococcum* sp. using a digestate from pig manure were able to accumulate carbohydrates under a nitrogen deficiency condition [52]. In this work, pig waste was subjected to anaerobic digestion, and the diluted digestate containing 49 mg L^−1^ N–NH_4_^+^ was used for cultivating this microalga under outdoor conditions. This source of N was depleted after 9 days of cultivation, and the initial total carbohydrate content of 20% dw increased up to 45.4% dw after 24 days. The authors concluded that this strain of *Chlorococcum* sp. has potential for bioethanol production using digestate of pig waste since the content of the reducing sugars in the biomass was 19.5% dw at the beginning of the cultivation period, and they increased up to 75% dw after 24 days.

More recently, Mata et al. [128] reported the cultivation of *Spirulina* sp. LEB 18 in the wastewater from a desalination process amended with 25% of Zarrouk’s medium. These authors observed a higher content of carbohydrates (52.29%), lipids (12.79%), and ash (2.69%) compared to the control assay (47.91; 7.59 and 1.29%, respectively).

### 4.5. Production of Bioethanol from Microalgal Biomass

The use of carbohydrate-rich microalgal biomass for bioethanol production has certain advantages compared to lignocellulosic materials since microalgal carbohydrates are mainly in the form of starch and cellulose (with the absence of lignin); thus, they are easier to convert to monosaccharides in a biorefinery process [129,130]. In addition, the use of CO_2_ as a carbon source for cultivation, accelerated growth, high yields obtained, and the fact that they do not compete for a large surface of the land for agricultural production make microalgae biomass an option with great potential to meet future demand for clean and sustainable energy [131].

Different species of microalgae, composed of varying percentages of total carbohydrates in their cells and cell wall, have been reported for bioethanol production. Ho et al. [132] cultured *Scenedesmus obliquus* CNW-N on a modified Detmer medium in batch-operated tubular photobioreactors (PBR) under outdoor conditions for one year with N deficiency and obtained carbohydrate productivities (composed of 70–80% glucose) that ranged from 47.3 ± 6.0 mg L^−1^d^−1^ (winter season) to 83.9 ± 9.9 mg L^−1^d^−1^ (summer season). A separate process of acid hydrolysis and fermentation was used to assess bioethanol production. The microalgal biomass was hydrolyzed with 2.0% sulfuric acid, and the hydrolysate was autoclaved (121 °C, 20 min) and then cooled (room temperature) and centrifuged (4 °C, 9000× *g* for 20 min). The hydrolysate pH was adjusted to 6.0 with CaCO_3_, and the solid fraction was then removed by centrifugation (10,000 rpm, 10 min). *Zymomonas mobilis* was used for the fermentation stage after being pre-cultivated at 30 °C. It was centrifuged at 10,000 rpm for 10 min and then inoculated to the hydrolysate of the microalgal biomass at an inoculum size of 10% (or optical density, OD600 = 2.0). Fermentation was carried out at 30 °C with an agitation of 150 rpm. The maximum bioethanol yield obtained was 0.205 (g g^−1^ dw biomass).

Onay [133] cultivated *Nannochloropsis gatidana* DEE003 in batch operated PBR under controlled conditions, at different ratios of municipal wastewater and filtered f/2 medium (0%, 30%, 60%, and 100% wastewater in filtered f/2 medium). The maximum carbohydrate content was obtained using 30% wastewater (17.7 ± 0.9%). After harvesting, biomass was lyophilized and then mixed with NaOH (1M) and autoclaved (121 °C, 30 min). After that, the hydrolysate was cooled (room temperature) and centrifugated at 4000× *g* for 10 min. *Saccharomyces cerevisiae* was used for bioethanol production. 3% (*v/v*) of the culture of *S. cerevisiae* was inoculated into the hydrolysate of microalgal biomass. The fermentation was carried out at 90 °C and 150 rpm for 48 h in anaerobic conditions. The bioethanol yield obtained in the treatment with 30% of wastewater was 94.3 ± 5.5 mg g^−1^ dw biomass.

Chandra et al. [134] used *Scenedesmus acuminatus*, grown in 11 N medium under controlled conditions operated in batch, and evaluated the effect of the deficiency of N–NO_3_^−1^ and P–PO_4_^−3^, the addition of Mg and lysine and the variation of pH in the medium on the carbohydrate accumulation, among other parameters, for bioethanol production. After harvesting, the biomass was dried and hydrolyzed with 2 N H_2_SO_4_ and then autoclaved. The hydrolysate pH was adjusted to 5.5. *S. cerevisiae* was inoculated into the filtered hydrolysate for fermentation, which was carried out at 30 °C and 200 rpm. The results obtained were a maximum carbohydrate and bioethanol productivity of 38.4 ± 2.4 mg L^−1^d^−1^ and 17.2 ± 1.2 mg L^−1^d^−1^, respectively, under a pH 9, with the addition of 0.5 g L^−1^ of lysine to the medium.

Recently, Constantino et al. [135] described the chemoenzymatic hydrolysis of freeze-dried biomass of *Chlorellla sorokiniana*, *Tetraselmis* sp. (Necton), and *Skeletonema* sp., and the subsequent fermentation of the hydrolysates with higher content of reducing sugars. The chemical hydrolysis was carried out using 4% *v/v* H_2_SO_4_ and autoclaving at 121 °C for 30 min. For the enzymatic saccharification, the authors used α-Amylase from *Aspergilus oryzae* (30 U/mg) and Amyloglucosidase from *Aspergillus niger* (300 U/mL). The enzymatic hydrolysis was performed after the hydrothermal acid pretreatment and a previous pH adjustment. *Saccharomyces cerevisiae* was used for bioethanol production after being cultivated in a liquid YEPD medium at 150 rpm and 30 °C. An inoculum of 10% (*v/v*) was used for fermentation experiments. The hydrolysates of *Chlorella sorokiniana* stood out with a bioethanol yield of 0.464 ± 0.013 g g^−1^ reducing sugar and a bioethanol productivity of 0.344 ± 0.02 g L^−1^ h^−1^.

Although bioethanol production from microalgae is considered to be a very promising strategy to mitigate some of the most important environmental problems such as climate change and biofuel production, it is also true that it still faces challenges related to its large-scale production with a high productivity and commercialization [136,137]. The selection of microalgae species with high biomass productivity and high carbohydrate content, their harvesting, pre-treatment, and efficient fermentation process represent some of the most demanding challenges for microalgae to become a competitive feedstock for bioethanol production [138].

## 5. Polysaccharides from Cyanobacteria for Developing New Bioactive Products

Many cyanobacteria are characterized by the presence, outside their outer cell membrane, of additional external structures of a polysaccharidic nature classified, according to their characteristics, as sheaths, capsules, and slimes. The sheath is a well-defined electron-dense thin layer that surrounds single cells, single filaments, or cell groups, reflecting their shape. The capsule is a gelatinous layer that is intimately associated with the cell surface and capable of absorbing large quantities of water. The capsule is characterized by sharp outlines and is structurally coherent to exclude particles. The slime is an amorphous, mucilaginous shroud, only loosely associated with the cell surface and capable of enclosing a large number of cells or filaments that may or may not be also surrounded by capsules. In the three cases, most exopolysaccharide (EPS)-producing cyanobacteria release into the culture medium during the growth or at the end of the linear phase of the growth, and large amounts of water-soluble polymeric material that was named released polysaccharides (RPS) [139]. Most of these polymers are characterized by an anionic nature, owing to the presence of uronic acids and/or other charged groups such as the sulphate or ketal-linked pyruvate groups [140].

Owing to the peculiar features of the polysaccharides produced by these microorganisms, EPS-producing cyanobacteria have been considered a promising resource of biopolymers of applied interest, and their exploitation to produce macromolecules suitable for specific industrial applications has been proposed in many studies. The specific features that have been considered of particular interest for the exploitation of cyanobacterial EPSs for developing new industrial products for biomedical applications are the following [140]:(i)Most of the polymers so far studied (up to now close to 200) are characterized by the presence in the macromolecule of a large number of different types of monosaccharides (in about 75% of the cases ranging from 6 up to 15), a feature rarely found in the EPS of other microorganisms. This feature makes it possible to have polymers with similar composition but organized in various structures and conformational forms, enlarging the chances of finding new products with useful properties;(ii)In most of the polymers are present uronic acids and sulphate groups, which are rare in other bacterial EPS and confer an anionic charge to the polymers. In addition to that, the presence of sulphate groups has been considered relevant for conferring antiviral properties to the polymers [141];(iii)The presence of peptides and deoxysugars, which confer hydrophobicity of some areas of the polymers;(iv)The frequent presence of uncommon sugars, such as acetylated or amine sugars, which are considered possibly involved in polymers’ biological activity.

Owing to these specific features, the industrial interest in EPSs of cyanobacterial origin has been rising, and a number of studies showed that some of the polymers so far characterized possess interesting biological activities.

A part of the international research has focused on the therapeutical applications of cyanobacterial polysaccharides, spanning a wide range of biological activities, namely antiviral, antibacterial, antioxidant, immunostimulatory, anti-inflammatory, antitumor, and also wound-healing (stimulating collagen synthesis) activities. A brief summary of these reported activities is given here, focusing on the literature specifically related to cyanobacterial EPS.

It was shown that the presence of sulphate groups confers antiviral properties to the sulfated exopolysaccharide (TK V3) produced by *Arthrospira platensis*, which was found to inhibit orthopoxvirus and other enveloped viruses [142]. Antiviral properties have also been shown for Nostoflan, an EPS produced by the terrestrial cyanobacterium *Nostoc flagelliforme* [143], as well as for the EPSs produced by other cyanobacteria [144].

Antibacterial activity is much more controversial, as studies conducted on *A. platensis* reported a different activity depending on the extracting solvent: ethanolic EPS extract showed no significant effect, while methanolic extracts showed bacteriostatic effects [145].

Regarding the use of EPS as antioxidants, the just mentioned *A. platensis* EPS ethanolic extracts showed a low antioxidant capacity, while methanolic extracts resulted in possessing a significantly higher antioxidant capacity [145]. It has to be stressed that most of the research about the antioxidant activity of EPS has been conducted on microalgal polysaccharides [146]. However, cyanobacterial EPS presents very interesting chemical properties which are generally considered relevant for radical scavenging, as the negative charges given and the complex tridimensional structure [147], this specific activity has not been well characterized yet.

Very recently, the highly sulfated (≈13% w/w) EPS produced by *Phormidium* sp. ETS05 showed to exert anti-inflammatory and pro-resolution activities in chemical and injury-induced zebrafish inflammation models, downregulating NF-κB signaling and reducing neutrophil recruitment, thus accelerating the clearance of these cells to recuperate tissue homeostasis [148]. The immunomodulation effect of sulphated EPSs produced by *Cyanothece* sp. PE 14 [149] or by *Cyanobacterium aponinum* [150] was also demonstrated in animal and human cell lines. *N. commune* extract resulted in being effective in downregulating IL-6 [151] and is, therefore, potentially useful for anti-allergic and wound-healing therapeutic purposes [152].

More recently, a number of studies showed interesting antitumor activities of the EPSs of cyanobacterial origin. Li et al. [153] reported the antitumor activity of the EPS produced by *Nostoc sphaeroides*. Ou et al. [154] showed that the EPS produced by the cyanobacterium *Aphanothece halophytica* (EPSAH) was capable of inducing apoptosis in HeLa cells. More recently, Flores et al. [155] showed that the EPS produced by a *Synechocystis ΔsigF* mutant is capable of decreasing the viability of melanoma, thyroid, and ovary carcinoma cells by inducing high levels of apoptosis.

The results reported above point out the potential of some of the EPSs produced by cyanobacteria for developing new products for biomedical application, but at the same time show the need for further studies for unveiling the molecular mechanisms of action of these macromolecules as well as their possible negative effects on healthy animal or human cells.

This brief overview of the research in the field of cyanobacterial EPS bioactivity puts into light the numerous potential applications of these polymers. However, their use in animal and clinical trials are yet to be explored as the high molecular weight of these molecules makes them complex to handle, which is also related to their rheological behavior [156].

## 6. Microalgae-Bioactive Compounds, A Natural Source of Potential Therapeutical and Health Promoting Agents: Insights from Innovative In Vivo Functional Studies

Microalgae are light energy-based biofactories that synthesize high-value bioactive compounds, being the more representative examples with health relevance: proteins, pigments, polyphenols, antioxidants, polyunsaturated fatty acids, vitamins, minerals, sterols, and polysaccharides. These bioactive compounds have multiple health properties as antiviral, anticancer, antioxidant, antidiabetic, antibacterial, antifungal, anti-inflammatory, neuro, cardiorespiratory and hepatoprotective, among others [7,10,157,158]. Within this frame, herein, recent functional studies in animal of microalgae-bioactive compounds with their remarkable health-promoting biological activities are summarized. These innovative microalgae bioactivities revisited here support the relevance that algal biomass has gained and are nowadays considered a valuable source of health-promoting agents with potential nutritional, biomedical, and therapeutical applications.

Furthermore, microalgae-bioactive compounds are potential health-promoting agents for the prevention and therapeutic application of relevant infectious, chronic, and degenerative diseases such as viral, fungal, bacterial infections, diabetes, metabolic, cardiorespiratory, cancer, inflammatory, neurodegenerative, among other pathologies.

### Algal Health-Promoting Agents

Microalgae-bioactive compounds have shown very important therapeutical bioactivities for the prevention or treatment of diseases as proven by functional studies in animals:Potential skin healing/antifibrotic agents:

A study performed with a rat model proved that when *Arthrospira platensis* was applied over skin wounds of Wistar rats, this microalga helped in the healing process. In addition to skin reparation, molecular expression patterns as indicators of the upregulation of angiogenic genes and downregulation of fibrotic genes were also observed [159]. Another study conducted with a rat model showed increased skin wound healing due to a diet with docosahexaenoic acid (DHA) from *Schizochytrium* sp., demonstrating its immunostimulant effect [160].

Promising liver health-promoters:

The administration of *Dunaliella salina* showed to revert liver disfunction, decreasing inflammation and oxidative stress by its anti-inflammatory and antioxidant properties, as it was observed in thioacetamide- (TAA-) induced hepatic encephalopathy rat model [161].

Natural immunostimulators:

Exopolysaccharides from *Porphyridium cruentum* (purpureum) administrated to shrimps *Litopenaeus vannamei* resulted in functioning as immunostimulators elevating the immune response of shrimps and protecting them from Vibrio infection [162]. Another functional study performed with Senegalese sole larvae treated with microalgae showed that *Nannochloropsis gaditana* and *Phaeodactylum tricornutum* could induce immune response [163]. In an intestinal inflammation murine model, it has been proven that *Arthrospira platensis* induced immunomodulatory effects [164]. A study realized with a murine model indicated that fucoxanthin could be used as a natural health-promoting agent as it showed an anti-inflammatory effect, facilitating the recovery of dextran sulfate sodium (DSS)-induced colitis mice [165].

Non-toxic anticancer agents:

An engineered *Chlorella vulgaris* was injected into a tumor-bearing mouse model showing to have antitumoral and anti-metastasis effects [166]. An in vivo rodent model showed that astaxanthin is safe and can be used as a nutraceutical and as an anticancer agent as it inhibits lung metastasis in mice [167].

Improvement of nutrients deficiency:

A functional study with a murine model indicated that *Nannochloropsis oceanica* could be considered an algal nutraceutical as its consumption increased hemoglobin values in anemic mice [168].

Prevention of type 2 Diabetes Mellitus development:

Microalgae antidiabetic agents are gaining relevance in the prevention and treatment of diabetes mellitus type 2 [169], as this disorder is considered to be the ninth worldwide leading cause of mortality, as 1 million deaths per year have been attributed to this metabolic disorder [170]. In a study performed with streptozotocin-induced diabetic rats, it was found that polysaccharides from *Porphyridium cruentum* had antihyperglycemic activity representing a potential natural antidiabetic agent [171]. In a murine model, it was demonstrated that consumption of n-3 fatty acids from microalgae could function as natural antidiabetic agents as an increment in the antioxidant capacity in adipose tissue of diabetic mice was observed [172].

Natural antioxidant agents:

These health-promoting agents could decrease tissue oxidation and cell damage, as demonstrated by a functional study performed with a rat model, which showed that the consumption of *Chlorella vulgaris* acted as an antioxidant agent that decreased skeletal muscle oxidative stress preventing cell damage [173]. A mice model indicated that carotenoids from *Scenedesmus obliquus* demonstrated that these pigments could be used as antioxidant agents, considering that the endogenous antioxidant defense system of mice was increased [174].

Herein, based on functional bioactivities, it has been illustrated how natural bioactive compounds considered high-added value products derived from microalgal biomass are gaining more impact as natural therapeutic agents useful for the prevention or treatment of diseases. Furthermore, their relevance as a valuable source of innovative natural agents with potential biomedical and therapeutical applications is highlighted.

## 7. Other Promising/Novel Trends

Microalgae-based biorefineries have been oriented towards the production of biomass of some valuable species, biofuels, high-added value products such as polysaccharides, pigments, and/or low-value products such as biofertilizers and biostimulants from the extracted biomass. However, future research should embrace promising and novel trends, which could improve the ratio of operation cost/economic benefit of such biorefineries. This section provides a short number of examples recently published.

Screening of microalgae looking for antimicrobial compounds has been extensively reviewed by Stirk and van Staden [175]. These authors recommended a thorough identification of potential strains quantifying the minimum inhibitory concentration (MIC) values. Antimicrobial activity was considered good if the MIC values were < 1 mg mL^−1^, moderate if MIC values were 1–8 mg mL^−1^, and weak with MIC > 8.0 mg mL^−1^. Their extensive analysis concluded that the Cyanobacteria and Chlorophyta were the phyla most promising. The challenge for future research is to cultivate and to scale up such species with higher potential and to include them in biorefineries to produce high-added value products.

On the other hand, a promising trend in the field of microalgae-based biorefineries is the use of pre-treated wastewater, anaerobic digestates, or crude wastewater for providing most of the nutrients required by the microalgae. In this manner, if the biorefinery is oriented toward a multi-product approach, the ratio of operation cost/economic benefit improves significantly. The challenges involved in the treatment of very different wastewater types with microalgae and the potential application of the produced biomass have been recently reviewed [176]. These authors concluded, among other relevant issues, that more detailed studies would be required to select appropriate strain(s) and, evaluate the long-term biomass productivity under fluctuating conditions and using large-scale outdoors reactors. In this regard, early work carried out with *Arthrospira maxima* cultivated in seawater supplemented with digestates from pig waste was performed in outdoor raceways (23.6 m^2^), operated semi-continuously for nearly 3 years under tropical conditions. The average annual productivity for semi-continuous cultures operating with depths of 0.10 m for winter and 0.15 and 0.25 m for the rest of the year was 11.8 g m^−2^ d^−1^. Biomass containing 48.9% of protein (ash-free dry weight), suitable as feed in aquaculture, was obtained [19]. Furthermore, a recent work reported the very high productivity of *Scenedesmus* sp. for treating raw domestic wastewater using large-scale outdoor raceways (11.8 m^3^ and 80 m^2^ of surface). The highest productivity found was 25.1 g m^−2^ d^−1^, when operating at a dilution rate of 0.2 d^−1^ in summer [177].

Currently, due to the significant exponential growth of the human population, the pollution of ecosystems is increasing exponentially. It is well known that a huge challenge is to decontaminate the pollution of air, water, soil, and food caused by human activities that have caused the appearance of health and environmental threats [178,179,180,181]. Therefore, the application of microalgae biomass obtained from biorefineries for bioremediation of wastewater and for its application as biofertilizers and biostimulants is becoming a new trend with the benefits of enriched soil and achieving an increment in food productivity in a sustainable, ecological, and health-friendly way.

Searching for novel fields, Cheng et al. [182] reviewed the use of marine microalgae strains in biorefineries, emphasizing the relevance of screening dominant microalgae together with their optimal culture conditions to obtain several valorization options of the algal biomass to ensure economic viability. It seems that marine microalgae are underexploited, and this constitutes a promising field to explore.

Another novel approach is the production of bioplastics in a microalgae-based biorefinery treating wastewater. Bioplastics are easily biodegradable, and some microalgae species have shown to accumulate poly (hydroxy alkanoate) esters (PHAs), among which poly (3-hydroxy butyrate) ester (PHB) is the most well-characterized biopolymer and have been found to accumulate at a concentration of 5.5–65% of dry biomass weight [183]. It has been extensively discussed that although this approach has much potential, many limiting factors are still to be overcome [184]. The first one is the requirement to maintain cyanobacteria-dominated cultures over prolonged periods. It is known that green microalgae are the major contaminants of cyanobacterial cultures treating wastewater. Thus, the authors recommended the maintenance of high N:P ratios (>32:1) and the utilization of waste streams with stable characteristics (e.g., diluted digestate) or low nutrient sources (e.g., secondary effluents).

Finally, current and future research should be embraced that follows the concept of the circular economy already discussed in previous sections and in Figure 2. This field is only emerging and still encounters numerous challenges in the case of microalgae-based biorefineries. A multi-product approach is quite advisable in terms of cost-benefit. However, the larger the number of products, the larger the investment and the need for counteracting all of the limitations listed in Table 3. Likewise, the environmental impact of such complex multi-product biorefineries needs to be carefully evaluated. In this regard, Ubaldo et al. [185] have recently reviewed various life cycle assessment studies and established future design choices for microalgal biorefineries. This is another very promising field that could help to properly enforce the concept of circular bioeconomy in relation to biorefineries. Furthermore, it has been recently reported that there is a lack of public policy adopted by regional entities, which could promote the adoption of policies for the implementation of the circular economy [186]. Thus, the field of circular economy implementation in the design and operation of microalgae-based biorefineries becomes a very promising trend to follow in the future.

The above-mentioned promising/novel trends offer multiple opportunities as well as numerous challenges. It is advisable to develop new research lines to overcome such limitations.

## 8. Conclusions

The microalgae-based biorefineries following the multi-product design and the circular economy approach are an important strategy to overcome the constraints when a single product is the final goal. New strains or new types of processes have been investigated worldwide, and this new information might lead to a more economically feasible large-scale production of high-added value products. Mixotrophic cultures using wastewater are becoming a strategic approach with successful results. Polysaccharides from cyanobacteria offer a new range of bioactive compounds. Innovative functional studies in animals have shown that microalgae-bioactive compounds have become a valuable emerging platform of natural health-promoting and medicinal agents with high therapeutical potential and relevant biotechnological and biomedical applications. Furthermore, promising/novel research lines, including the screening of microalgae for antimicrobial compounds, the use of wastewater for the production of several new products (i.e., bioplastics), and the application of the Life Cycle Assessment tool to follow the circular economy strategy, offer a wide range of challenges and opportunities for overcoming limitations, especially for scaling-up processes and achieving a convenient cost/benefit ratio.

## Figures and Tables

**Figure 1 biology-11-01146-f001:**
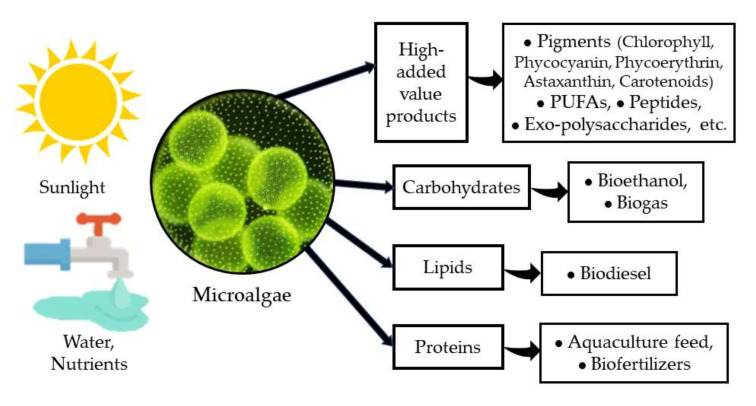
Microalgae and their products.

**Figure 2 biology-11-01146-f002:**
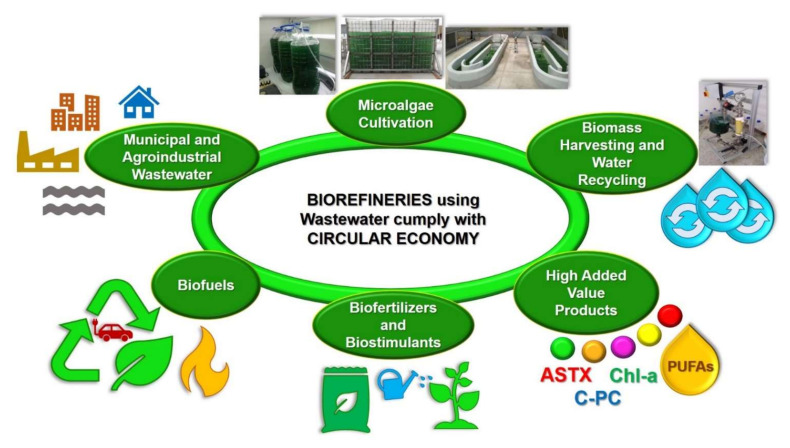
Microalgae-based biorefineries using and treating wastewater, follow the Circular Economy concept.

**Figure 3 biology-11-01146-f003:**
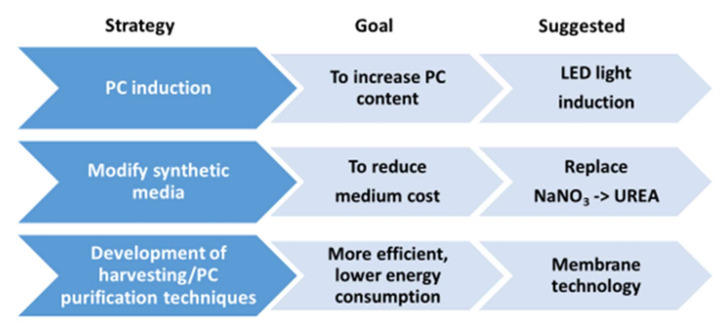
Potential strategies suggested to achieve PC production in a production facility (adapted from [40]).

**Table 1 biology-11-01146-t001:** Pigments production using photoautotrophic microalgae cultures.

Pigment	Microalgae/Cyanobacteria Species	Country	Bioprocess Operational Strategy	Pigment Content/Productivity	Reference
Phycocyanin	*Nostoc* *commune* TUBT05 *Oscillatoria okeni* TISTR8549	Thailand	Batch	543.7 ± 28.78 mg g^−1^	[38]
	*Leptolyngbya* sp. QUCCCM56	Qatar	Batch	169.9 ± 3.6 mg g^−1^	[39]
	*Arthrospira (Spirulina) maxima* LJGR 1	Mexico	Pilot-plant level	95–135 mg g^−1^	[40]
Astaxanthin	*Haematococcus pluvialis*	China	Batch	140 mg m^−2^ d^−1^	[41]
	*Chromochloris zofingiensis*	China	Batch	38.9 mg L^−1^	[42]
	*Chlorella sorokiniana* IBRC-M50023	Iran	Batch	24.2 ± 0.1 mg L^−1^	[43]
	*Haematococcus pluvialis* NIES-144	Republic of Korea	Batch	10.3 ± 0.4 mg L^−1^	[44]
Lutein	*Chlorella sorokiniana* SAG 211/8K	The Netherlands	Batch	7.3 ± 0.5 mg g^−1^	[45]
	*Chlorella sorokiniana* MB-I-M12	Taiwan	Semi-continuous	4.46 mg L^−1^ d^−1^	[46]
	*Chlorella* sp. GY-H4	China	Semi-continuous	38.5–63 mg L^−1^	[47]
	*Koliella antárctica* *Tetraselmis chui*	Italy	Batch	0.27–2.05mg g^−1^ 0.29–2.2 mg g^−1^	[48]
	*Chlorella* sp. *AE10*	China	Batch	4.4 mg L d^−1^ (red light) 9.58 mg g^−1^ (blue light)	[49]
Allophycocyanin	*Arthrospira* *platensis*	Brazil	Fed batch	13.3 mg g^−1^	[50]
Phycoerytrin				4.1 mg g^−1^	
Violaxanthin	*Nannochloropsis gatidana*	Italy	Batch	0.24–2.07 mg g^−1^	[48]
Chlorophyll	*Chlorella sorokiniana* SAG 211/8K	The Netherlands	Batch	35.4 1.7 mg g^−1^	[45]
	*Nannochloropsis gatidana*	Italy	Batch	1.8–13.8 mg g^−1^	[48]

**Table 2 biology-11-01146-t002:** Pigments produced by mixotrophic cultures of microalgae using wastewater.

Pigment	Microalgae Specie	Wastewater Type	Bioprocess Operational Strategy	Pigment Content/Productivity	Reference
Phycocyanin	*Arthrospira platensis*	Palm oil municipal effluent	Fed batch	120.13 ± 1.1 mg g^−1^	[56]
	*Nostoc* sp. *Arthrospira platensis* *Porphyridium purpureum*	Industrial	Batch	103 mg g^−1^	[64]
Astaxanthin	*Haematococcus pluvialis*	Potato juice	Batch	14.7–27.9 mg g^−1^	[62]
Lutein	*Chlorella sorokiniana*	Industrial and municipal	Batch/Continuous	1.03 mg g^−1^	[59]
	*Chlorella* sp. GY-H4	Diluted food waste hydrolysate	Semi-continuous	44 mg L^−1^ (with an initial glucose concentration of 10 g L^−1^) 63 mg L^−1^ (with an initial glucose concentration of 20 g L^−1^)	[47]
β-carotene	Desmodesmus spp.	Industrial	Batch	* 0.6473 mg g^−1^	[60]
	*Dunaliella salina*			* 0.7435 mg g^−1^	
	*Nannochloropsis limnetica*			* 0.7435 mg g^−1^	
	*Chlorella sorokiniana*			* 1.039 mg g^−1^	
	*Nephroselmis* sp.	Industrial	Batch	* 0.188 mg g^−1^	[61]
	*Dunaliella* sp. FACHB-558	Poultry litter	Batch	7.26 mg L^−1^	[63]
Violaxanthin	*Desmodesmus* spp.	Industrial	Batch	* 0.0546 mg g^−1^	[60]
	*Dunaliella salina*			* 0.08301 mg g^−1^	
	*Nannochloropsis limnetica*			* 1.228 mg g^−1^	
	*Nannochloropsis salina*			* 1.679 mg g^−1^	
Allophycocyanin	*Nostoc* sp. *Arthrospira platensis* *Porphyridium purpureum*	Industrial	Batch	57 mg g^−1^	[64]
Phycoerytrin				30 mg g^−1^	
Chlorophyll (Chl)	*Botryococcus braunii*	Municipal	Batch	(Chl-a) 32.21 µg ml^−1^ (Chl-b) 50.51 µg ml^−1^	[57]
	*Chlorella sorokiniana*	Industrial and municipal	Batch/Continuous	18.01 mg L^−1^ d^−1^	[59]

* data obtained from own calculations with author’s information.

**Table 3 biology-11-01146-t003:** Advantages, limitations, and opportunities for future research related to mixothrophic cultures in Microalgae-based biorefineries.

**Advantages** The circular economy concept is implemented [69,70,71,72]; They can be used for treating wastewater and producing microalgae biomass [18,19,20,21,25]; A high carbon fixation rate by microalgae (1.83 kg CO_2_ kg−1 of biomass) mitigates climate change [73]; Nutrients from wastewater allow lower production costs [25,53]; Clean Water use is reduced significantly [74]; It is possible to produce high-added value products (HAVP) [73,75]; It is advisable to design a multi-product biorefinery [25,73]; The combination of HAVP with lower-value products is advisable [25,51]; The use of stress conditions to induce carbohydrate accumulation allows the production of bioethanol, polysaccharide bioactive compounds, and other valuable novel biopolymers such as bioplastics [6,76,77]; Microalgae cultivation at a large scale is well established for some species [78]; Valuable species such as Arthrospira sp. have been shown to grow in some pre-treated agroindustrial wastewater [18,19]. More recently, many other efforts of cultivation of other microalgae species using digestates or diluted pig slurry have been reported [26,27,65]. **Limitations and Opportunities for future research** It is not always feasible to reach the zero-waste goal [69,79]; Not all types of wastewaters can be treated and simultaneously used for biomass production [80,81]; Some aggressive wastewater containing toxic compounds or high organic matter content require pre-treatment [80,82]; If the wastewater requires pre-treatment, additional clean water is required [79,82]; Recycling of water after harvesting the biomass or after anaerobic digestion of it is limited due to the presence of remaining macro and micronutrients, unassimilated ions, pH, concentration, and type of toxic elements [83]; Some HAVP are not acceptable in the market if wastewater was used [40]; The use of microalgae biomass as biofertilizer or biostimulant for agricultural crops is an emergent field [71]; Some promising microalgae species are still under study at lab scale [78]; Most species cannot grow axenically in mixotrophic cultures [84].

**Table 4 biology-11-01146-t004:** Major macromolecules percentage in the biomass of various microalgae cultivated under specific conditions.

Microalgae Species	Culture Conditions	Macromolecule (% dry *w*/*w*)			Reference
		Carbohydrate	Protein	Lipid	
*N. oleabundans*	Nutrient deficiency induced using various percentages of anaerobic digestates of vinasse and the addition of NaHCO_3_ (under controlled conditions) and three different culture media in flat plate photobioreactors (under uncontrolled conditions) resulted in lipid accumulation	-	-	18–39	[26]
*Spirulina platensis* LEB 52	Nutrient deficiency induced by using a Zarrouk medium diluted to 20% amended with 2.5% of two different residues from the ultra and nanofiltration of whey protein.	45–58	51	-	[112]
*Arthrospira* sp. ZJWST	Raceways pond cultivation under nutrient limitation induced by the use of digested piggery wastewater pre-treated with a membrane bioreactor and ozonation amended with NaHCO_3_.	-	59	-	[113]
*Chlorococcum* sp.	Batch culture under nitrogen starvation using a digestate from pig manure as a nutrient source under indoor and outdoor conditions.	42–45	16–20	11–13	[52]
*Scenedesmus* sp.	Nutrient limitation induced by different dilution rates, centrate (from the anaerobic digestion of urban wastewater) percentages, and culture depths.	-	-	10	[21]
*Chlorella vulgaris*	Nutrient deficiency induced by different dilution percentages by mixing natural lake water, aquaculture wastewater (AW) and pulp wastewater (PW). The experimental units with the highest growth were selected for microalgae culture in AW and PW without nutrient addition.	18–19	44–46	44–50	[114]
*Chlorella vulgaris*	Different initial microalgae density (20–65 mg L^−1^) and nitrogen starvation during 14 days of microalgae culture.	-	-	32	[115]
*Chlorella vulgaris*	Co-cultivation of microalgae with filamentous fungi was used for molasses wastewater treatment. This strategy was carried out by inoculating microalgae and fungi in wastewater under the optimized conditions (temperature and inoculation ratio on biomass production).	-	62	22	[116]
*Tribonema* sp.	The algae cultivation was performed using open vertical tubular PBR with an initial culture density of 0.2 g L^−1^. Nutrient starvation was induced by the differences in the chemical composition of petrochemical wastewater effluents (used directly without further treatment).	55	>20	34–36	[117]
*T. obliquus* KNUA019 *A. quadricellulare* KNUA020 *Desmodesmus* sp. KNUA024 *Pseudopediastrum* sp. KNUA039	Microalgae strains pellets were suspended with filtered municipal wastewater and inoculated 10% into 1 L sterile flasks containing filtered treated primary settled wastewater for 11 days.	24–28	32–40	16–26	[118]
*Chlamydomonas* sp. TRC-1	Nutrient starvation was induced by the differences in the chemical composition of effluents (GB-11 medium and textile effluents).	19	52	11	[119]
*Chlorella vulgaris* *Chlorella sorokiniana* UKM3 *Chlamydomonas* sp. UKM6 *Scenedesmus* sp. UKM9	Nitrogen depletion through acclimatization and continuous light exposure in anaerobic digested palm oil mill effluent (POME). Microalgae were cultured in anaerobic digested POME without any dilution or additional substrate.			21–50	[120]

## Data Availability

Not applicable.

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
