# Peer review of "Microalgae-Based Biorefineries: Challenges and Future Trends to Produce Carbohydrate Enriched Biomass, High-Added Value Products and Bioactive Compounds"

_biology, 2022, doi:10.3390/biology11081146_

Round 1

Reviewer 1 Report

The manuscript presents a very interesting review of the recent research about high added value components of microalgae and strategy for increasing the carbohydrate content of microalgae cultures. The document includes very interesting tables about microalgae and pigments production. In opinion of this reviewer, the article is adequate for publication in its current form. However, some recommendations/suggestions are provided below to improve the clarity of the article:

-      -    The review focuses on valorization of 1) the carbohydrate fraction of microalgae 2) high added value components. The recovery of other important components as proteins and lipids is not analysed in this work. Probably, this approach should be clearly stated and justified in the title, abstract, keywords and through the text.

-       -    Usually, proteins are the major component of microalgae biomass grown in wastewater treatment photobioreactors. The interest of applying operational strategies to transform proteins into carbohydrates should be justified, in terms of economic advantages and viability of the processes.

-         -  The concerns and limitations of applications of biomass grown in wastewater should be also advised. Specially in the summary and abstract, but also through the text, some readers without expertise in this field could understand the use of wastewater as cheap culture medium for producing microalgae bioactive compounds.

-          - Page 2, lines 47 and 48, this list of macro components can be confusing, polysaccharides are a type of carbohydrates.

-          - Figure 2: The idea of biorefineries of integral biomass valorization is not clearly shown in this figure. Biomass seems to provide only high added-value products. Additionally, biofuels appear from the central text, not from biomass.

-          - Page 6, line 150, please provide the units of 3.72 high purity

-          - Page 6, line 168-179, other biorefinery alternatives for valorization of microalgal biomass grown in wastewater should be included in this section. Also, the concerns of using wastewater (mainly domestic) for producing high added value products should be mentioned.

-          - Table 3. Please, the authors would include some discussion about the minimal concentration values for a viable process. What is the difference between domestic and municipal wastewater?

-          - Page 4. The interest of enriching biomass in carbohydrates should be adequately justified.

-         -  Page 9, lines 245-246: “In fact, at large-scale, the most important target is to achieve high carbohydrate productivities not high carbohydrate contents”. Please, ¿could you provide some example or data for this sentence?.  Are the nitrogen depletion starvation methods suggested to increase the carbohydrate concentration of microalgae in this section in agreement with this sentence?. Please, include the analysis of carbohydrate productivities when discussing the starvation experiments through the text.

-          - Page 9, lines 278-280, page 10 line 281: “In stage 2 of the culture, when both the concentration of CO2 applied and nitrogen removed from the medium decreased, the carbohydrate content in the microalgae increased…….after 4 days of N starvation”. Please, provide additional information and data to clarify this sentence. What means CO2 applications at 0.04%?

-          - Page 10, lines 289 and 292, please, use superscript for 10^6 number of cells.

-          - Page 10, line 291, please provide comparable data at different ICD (e.g., starch content)

-          - Page 11, page 353-355. Please provide the most usual ranges of macrocomponent compositions in microalgae biomass grown in wastewater. I recommend the authors to provide a table with this information.

-          - Page 12, lines 399-419. The ethanol yields provided in examples of this paragraph are really, high, almost stoichiometric values of the total carbohydrate content of biomass. These so high yields would require very high monosaccharides recovery yields in the hydrolysis step, no degradation and very high fermentation yields of all the recovered monosaccharides. Please, provide information about the hydrolysis conditions and the microorganisms used for the fermentation stage in these examples.

-          - Page 16, line 607, please provide the data of height in the raceway

-          - Page 16, lines 608-609, please revise the sentence about composition of biomass, it seems incomplete.

Reviewer 2 Report

Dear Authors,

The highlight, challenges and future trends were discussed related to the microalgae-based biorefineries including both, phototrophic and  mixotrophic cultures treating wastewater and the recovery of biomass as a source of valuable mac-30 romolecules and high-added value products.

In Table 1, the world production of Arthrospira is more than 10,000 tons/year. The production information should be mentioned. They were mainly from Ref. 7 while it was published in 2019. The data are too old. 

Some information in Table is too low. For example, the lutein content is only 0.0364 mg/g. If the data are right, why did the authors select this reference? The lutein content is easily higher than 5 mg/g. 

Biorefinery is the sustainable processing of biomass into a spectrum of marketable products (food, feed, materials, chemicals) and energy (fuels, power, heat). It is not for the production of only one product. Biorefinery is the multiple-product development. The authors should discuss the concept in a right way. There are lots of samples of microalgae process development by  biorefinery.

Li D, Yuan Y, Cheng D, Zhao Q. Effect of light quality on growth rate, carbohydrate accumulation, fatty acid profile and lutein biosynthesis of Chlorella sp. AE10. Bioresour Technol. 2019;291:121783.

Wang X, Zhang MM, Sun Z, Liu SF, Qin ZH, Mou JH, Zhou ZG, Lin CSK. Sustainable lipid and lutein production from Chlorella mixotrophic fermentation by food waste hydrolysate. J Hazard Mater. 2020 Dec 5;400:123258.

Reviewer 3 Report

A well written and organized manuscript.

I have following comments

1. Use of water in microalgae cultivation in very intensive and its also desired for nutrients reuse. Authors did not discussed it. I will recommend to be brief in nutrients effects section and discuss challenges with reuse of water in microalgae cultivation.

Following articles might be helpful

https://www.sciencedirect.com/science/article/pii/S0960852414015727

https://www.sciencedirect.com/science/article/pii/S2211926414001398

https://onlinelibrary.wiley.com/doi/full/10.1111/gcbb.12822

3. In wastewater treatment using microalgae, two stage cultivation technique is very promising as highlighted in the following article and many more. 

https://www.sciencedirect.com/science/article/pii/S0960852413000576

Reviewer 4 Report

General Comment:  The review is to discuss challenges and future trends related to the microalgae, the recovery of biomass as a source of valuable macromolecules and high-added value products. The therapeutic properties of some microalgae-bioactive compounds are also discussed. Novel trends such as the screening of species for antimicrobial compounds. The paper is well written, just minor comments I describe below.

 P8, L197. It seems to me that the light intensity part is only left as a compromise but that it should be dealt with in greater depth.

 P8, L215 and others lines. The authors mentioned several types of culture media, none is referenced

 P9, L232. This meaning of the acronyms must go before.

 P11, L362-367. Lack of reference to the information presented

Round 2

Reviewer 2 Report

Dear Authors,

I have no further comments.